# Targeting TLR2/Rac1/cdc42/JNK Pathway to Reveal That Ruxolitinib Promotes Thrombocytopoiesis

**DOI:** 10.3390/ijms232416137

**Published:** 2022-12-17

**Authors:** Shuo Yang, Xiaoqin Tang, Long Wang, Chengyang Ni, Yuesong Wu, Ling Zhou, Yueying Zeng, Chunling Zhao, Anguo Wu, Qiaozhi Wang, Xiyan Xu, Yiwei Wang, Rong Chen, Xiao Zhang, Lile Zou, Xinwu Huang, Jianming Wu

**Affiliations:** 1School of Pharmacy, Southwest Medical University, Luzhou 646000, China; 2School of Basic Medical Sciences, Southwest Medical University, Luzhou 646000, China; 3Education Ministry Key Laboratory of Medical Electrophysiology, Southwest Medical University, Luzhou 646000, China

**Keywords:** rusolitinib, MK, thrombocytopoiesis, radiation, TLR2/Rac1/cdc42/JNK

## Abstract

Background: Thrombocytopenia has long been considered an important complication of chemotherapy and radiotherapy, which severely limits the effectiveness of cancer treatment and the overall survival of patients. However, clinical treatment options are extremely limited so far. Ruxolitinib is a potential candidate. Methods: The impact of ruxolitinib on the differentiation and maturation of K562 and Meg-01 cells megakaryocytes (MKs) was examined by flow cytometry, Giemsa and Phalloidin staining. A mouse model of radiation-injured thrombocytopenia (RIT) was employed to evaluate the action of ruxolitinib on thrombocytopoiesis. Network pharmacology, molecular docking, drug affinity responsive target stability assay (DARTS), RNA sequencing, protein blotting and immunofluorescence analysis were applied to explore the targets and mechanisms of action of ruxolitinib. Results: Ruxolitinib can stimulate MK differentiation and maturation in a dose-dependent manner and accelerates recovery of MKs and thrombocytopoiesis in RIT mice. Biological targeting analysis showed that ruxolitinib binds directly to Toll Like Receptor 2 (TLR2) to activate Rac1/cdc42/JNK, and this action was shown to be blocked by C29, a specific inhibitor of TLR2. Conclusions: Ruxolitinib was first identified to facilitate MK differentiation and thrombocytopoiesis, which may alleviate RIT. The potential mechanism of ruxolitinib was to promote MK differentiation via activating the Rac1/cdc42/JNK pathway through binding to TLR2.

## 1. Introduction

Radiotherapy is the mainstay of cancer treatment at present. Although radiation mainly targets malignant tissues, it also damages the surrounding normal tissues and organs [1]. Acute radiation syndrome (ARS) affecting multiple organ systems is caused by short-term exposure of the whole or large parts of the body to high doses of radiation and may present with hematopoietic subsyndrome, gastrointestinal subsyndrome, and neurovascular subsyndrome [2]. Because blood-forming cells are extremely sensitive to radiation, even a few grays can cause significant damage to bone marrow, which results in a rapid depletion of white blood cells, platelets, and reticulocytes and may lead to serious complications such as anemia, bleeding, infection, and immune dysfunction [3]. Additionally, it has been noted that platelet count, more so than any other hematologic marker, is more closely related to survival rate following total body irradiation [4]. Thrombocytopenia (TP) is defined as the number of whole blood platelets below 150 × 10^9^/L [5]. Platelet transfusion, corticosteroids, platelet growth promoting factor, platelet stimulating hormone receptor agonist, and thrombopoietin are currently used in clinical trials. However, a number of side effects, such as an increased recurrence rate, severe side effects, and increased drug resistance, have prompted effective alternative drugs for TP treatment.

Platelets are important regulators of bleeding, infection, thrombosis, and inflammation [6]. A platelet is a tiny anucleate cell that is the terminal product of the maturation and differentiation of MKs [7]. MK maturation and the release of functioning platelets into the bloodstream are referred to as thrombopoiesis [8]. Hematopoietic stem cells in the bone marrow undergo continual and intricate biological processes to create MKs. MKs go through an endomitotic process in which the DNA replicates continuously but the cytoplasm does not divide, resulting in mature polyploid cells [9]. Under the influence of blood flow shear forces, mature MKs release proplatelets into the blood to become functional platelets [10]. Important MK surface-specific antigens CD41, CD61, and CD42b are gradually expressed. In this process, a dynamic, well-organized cytoskeleton of tubulin and actin is essential for MK development and proplatelet extension [11]. Small G proteins of the Rho family, including members RhoA, Rac1, and cdc42, are important actin cytoskeleton regulators that control cell spreading, motility, and growth and have been shown to play important roles in platelet contraction, activation, secretion, aggregation, and thrombus stability of platelets [12,13]. Downstream of membrane receptors, Rac1 and cdc42 are considered potential effectors that can induce cell survival or death depending on the cellular situation [14]. They also mediate the activity of various kinases, such as JNK and p38/MAPK, which also affect cell growth, differentiation, survival, and apoptosis [15]. JNK is an important branch of the MAPK pathway and has been reported to play a very important role in MK differentiation, platelet production, and platelet function [16]. Once JNK is activated by various upstream players, it translocates to the nucleus and phosphorylates various transcription factors. Activator protein-1 (AP-1) is considered to be the key transcription factor affected by JNK [17]. It has been reported that in circulating blood, exposure to shear flow leads to activation of JNK and upregulation of AP-1 transcripts (ATF4, JUNB, JUN, FOSB, FOS, and JUND) in MK, thus promoting MK maturation and platelet biogenesis [16]. GATA-1, FOG-1, RUNX1, TAL-1, FLI-1, and NF-E2 are key transcription factors that regulate MK differentiation and platelet production [18]. Any abnormality in the development and maturation of MKs and platelet release may lead to platelet problems.

Ruxolitinib is a selective JAK1/2 inhibitor with moderate tyrosine kinase 2 inhibitory activity (Tyk2). It has been approved by the Food and Drug Administration for the treatment of intermediate- and high-risk primary myelofibrosis and polycythemia vera patients who have not responded or are intolerant to hydroxyurea [19]. In clinical treatment, ruxolitinib is also commonly used in nonmyeloproliferative diseases, including irritable bowel disease, rheumatoid arthritis, psoriatic skin lesions, and graft-versus-host disease [20]. Although it has been reported that ruxolitinib can treat leukemia in the hematologic disease category, the ability to recover from radiation-induced thrombocytopenia has not been explored.

Ruxolitinib appears to be an excellent candidate for the treatment of radiation-induced thrombocytopenia. In the current study, we demonstrated that ruxolitinib has the ability to promote MK maturation and platelet production by in vitro identification of activity. In vivo studies showed that ruxolitinib significantly promoted platelet recovery in thrombocytopenic mice. Mechanistically, ruxolitinib significantly promoted MK maturation by targeting TLR2, thereby regulating the Rac1/cdc42/JNK signaling pathway. In conclusion, our study provides new pharmacological insights into ruxolitinib. The multiple efficacy of ruxolitinib in the recovery process is crucial, which makes ruxolitinib a clinical candidate for the treatment of radiation-induced thrombocytopenia.

## 2. Results

### 2.1. Safe Concentration of Ruxolitinib for the Treatment of K562 and Meg-01 Cells

We evaluated the activity of K562 and Meg-01 cells treated with ruxolitinib for the specified period to establish the ideal ruxolitinib concentration for MK differentiation in vitro. In contrast to the control group, the proliferation of K562 and Meg-01 cells was inhibited by ruxolitinib treatment (Figure 1A,B). We could find that at day 3, K562 and Meg-01 treated with 20 μM ruxolitinib exhibited a decrease in cell proliferation rates, and at day 5, Meg-01 cells exhibited inhibition of proliferation at only 5 μM, and K562 cells exhibited significant inhibition of proliferation at 10 μM. This may be the result of cell differentiation [21]. LDH release is considered an important indicator of cell membrane integrity and is widely used for cytotoxicity testing [22]. There was no difference in the total LDH between the control group and the drug intervention group, indicating that 5, 10, and 20 μM ruxolitinib was within a safe concentration range for MK differentiation (Figure 1C,D). Therefore, for the in vitro study, we used drug doses of 5, 10, and 20 μM. Notably, PMA, which is known to induce MK differentiation, was selected as a positive control in the current study [23]. We found that microscopic images of treated cells showed that K562 and Meg-01 cells treated with ruxolitinib had a larger diameter and a significant increase in cell size compared to cells in the untreated group (Figure 1E). One possible explanation is that ruxolitinib inhibits MK proliferation and promotes MK differentiation.

### 2.2. Ruxolitinib Induces Typical MK Differentiation

CD41 and CD42b are specific surface antigens of MKs [24]. The effect of ruxolitinib at different concentrations (5, 10, and 20 μM) on the co-expression of CD41/CD42b in K562 and Meg-01 cells was detected by flow cytometry. The results showed that ruxolitinib significantly promoted CD41/CD42b expression in K562 and Meg-01 cells in a concentration-dependent manner within a safe range (Figure 2A,B). The maturation of MKs was accompanied by changes in cell morphology, polyploidization of nuclei, and formation of the demarcation membrane system (DMS) [25]. Giemsa staining showed that the increased nuclear-cytoplasmic ratio of cells, deep staining of the multilobal nucleus, and the development of differentiation. Cells treated with ruxolitinib showed polyploidization of MKs consistent with intranuclear mitosis, most of which were clearly distinguishable as 8N or 16N ploidy (Figure 2C). Cytoskeleton actin was visualized with rhodamine-labeled phalloidin staining, and nuclear polyploidy was observed by DAPI staining. On the 5th day of ruxolitinib intervention in K562 and Meg-01 cells, it was clearly observed that the number of multilobar nuclear cells increased, the cytoplasm of cells obviously expanded, and F-actin aggregation was induced, whereas it was rarely seen in the control group (Figure 2D). In conclusion, the above data suggest that ruxolitinib promotes the maturation of MKs in a dose-dependent manner within a safe concentration range.

### 2.3. Ruxolitinib Promotes Platelet Recovery in Irradiated Mice

Given the significant pro-MK differentiation effect of ruxolitinib in *vitro*, we evaluated whether it has a therapeutic effect in RIT mice. KM mice were subjected to systemic radiation at a single dose of 4.0 Gy, followed by intraperitoneal injection of saline in the control and model groups, with ruxolitinib (10 mg/kg) intervention in the experimental group and rhTPO (recombinant human thrombopoietin, 1500 U/kg) intervention in the positive control group at the same time each day (Figure 3A). The peripheral blood data showed a rapid and significant decrease in leukocyte levels after irradiation in day 0, which indicates the success of the RIT mice. With the extension of administration time, the level of leukocytes recovered slowly, but there was no difference between the experimental group and the model group (Figure 3B). There was no significant difference in erythrocyte levels between the ruxolitinib-treated, rhTPO-treated, and model groups at any time of the assay (Figure 3C). Although the peripheral platelet count of mice in the irradiated group dropped to the lowest on day 7, the peripheral platelet count of mice treated with ruxolitinib and rhTPO was also significantly higher than that of the model group, and it steadily and effectively promoted the recovery of platelet production in IR mice afterwards (Figure 3D,E). These results suggest that the restorative effect of ruxolitinib on peripheral blood cells in radiation-injured mice is limited to platelet-forming cells, but it does not reduce the cytotoxic effect of radiation on leukocytes, emphasizing the specificity for platelet-forming cells.

### 2.4. Ruxolitinib Rescues Bone Marrow MKs after Radiation Injury

IR-induced bone marrow suppression is a major contributor to hematopoietic injury and is characterized by loss of BM cell structure [26]. IR mice exhibit severe disruption of the BM and vascular system, leading to massive ablation of BM nucleated cells (BMNCs) [27]. The recovery of BM cell structure indicated that stem cells were preserved and stimulated. To determine whether the changes detected in peripheral blood are consistent with the level of megakaryocytes in the bone marrow, we carried out bone histological examination on mice on the 12th day of treatment. As expected, H&E staining showed that the number of MKs in the BM was much higher in the ruxolitinib- and rhTPO-treated groups than in the model group (Figure 4A,B), indicating that the recovery of bone marrow MKs was promoted after ruxolitinib administration. Next, we analyzed the cell population of MK-group cells (CD41) enriched in the bone marrow. We examined the cell populations of c-Kit^+^CD41^−^ (hematopoietic progenitor cells), c-Kit^+^CD41^+^ (megakaryocytic progenitor cells), and c-Kit^−^CD41^+^ (MKs). The results showed (Figure 4C,D) that there was no difference in the proportion of c-Kit^+^CD41^−^ cells in each group, while the c-Kit^+^CD41^+^ and c-Kit^−^CD41^+^ proportions were significantly higher in the ruxolitinib-treated and rhTPO groups than in the model group. This could be explained by the fact that the bone marrow cells of mice treated with ruxolitinib were transferred from immature c-Kit^+^CD41^−^ cells to co-expressed c-Kit^+^CD41^+^ and mature c-Kit^−^CD41^+^ cells. In addition, the expression of the MK markers CD41 and CD61 in BM cells was detected by flow cytometry. At the time point of the experiment, there was a modest increase in cellular data for these groups in the bone marrow of mice treated with ruxolitinib and rhTPO compared to the control group of unirradiated mice (Figure 4E,F). Differences in the fractions of hematopoietic surface markers CD41^+^ and CD61^+^ cells suggest that ruxolitinib triggers the maturation of MK cell formation at different stages of megakaryopoiesis. The increased number of MKs in the ruxolitinib-treated group may result from the restoration of hematopoietic progenitor cells by ruxolitinib and stimulation of the differentiation of hematopoietic progenitor cells toward MKs.

### 2.5. Ruxolitinib Restores Splenic Hematopoiesis

The spleen can undergo extramedullary hematopoiesis (EMH) triggered by physiological stress or disease and can be used as an alternative tissue site for bone marrow hematopoiesis [28]. Therefore, we studied the effect of ruxolitinib on splenic MK production in mice after radiation. Radiation damage caused mice to exhibit extensive atrophy of the splenic body [29]. First, the ratio of spleen weight to body weight was not significantly different in the ruxolitinib-treated versus rhTPO-treated mice compared to the model group (Figure 5B), and H&E staining results also showed less splenic atrophy in the ruxolitinib-treated and rhTPO-treated mice, accompanied by reconstruction of splenic sinus structures and a corresponding increase in MK numbers (Figure 5A,C), suggesting that spleen promotes megakaryopoiesis. Then, as with bone marrow cells, we detected the expression of the MK markers CD41 and CD61 in splenic cells by flow cytometry, and ruxolitinib significantly increased the ratio of CD41^+^ cells and CD41^+^/CD61^+^ cells in radiated mice (Figure 5D,E).

### 2.6. Ruxolitinib Promotes the Number and Functional Recovery of Peripheral Blood Platlets

To confirm there are no ineffective platelet production, we performed CD41/CD61 flow cytometry analysis of cells in peripheral blood (Figure 6A,B). The results showed that the CD41^+^/CD61^+^ ratio in peripheral blood cells was significantly higher in the ruxolitinib-treated and rhTPO groups, consistent with bone marrow cells and spleen cells, which indicated effective hematopoiesis in bone marrow and spleen.

Platelet activation and coagulation are involved in hemostasis and platelet production [30]. One of the major biomarkers of platelet activation is P-selectin (CD62P), a glycoprotein present in resting platelet alpha granules [31]. Activated platelets dynamically change their shape and release alpha-granule contents and secrete a variety of cytokines, chemokines, and growth factors. The expression of CD62P on the platelet surface reflects cell degranulation; therefore, measuring its levels is a useful tool to monitor the status of platelet activation in vitro and in vivo [32]. A CD41/CD62P flow assay was performed on peripheral blood cells, and CD41^+^/CD62P^+^ expression was significantly elevated in peripheral blood cells of mice in the ruxolitinib and rhTPO treatment groups compared to mice in the model group (Figure 6C,D). The platelet response to physiological agonists is a measure of circulating platelet availability and depletion [31]. Based on flow cytometry measurements, our results of CD62P for peripheral blood washed platelets showed significantly higher basal CD62P expression on the platelet surface in the ruxolitinib- and rhTPO-treated mice compared to the model group (Figure 6E,F). Moreover, in the CD62P assay after stimulation with ADP, platelet reactivity was significantly enhanced in the mice in the ruxolitinib and rhTPO treatment groups (Figure 6G). A tail bleeding model was used to detect the hemostatic function of platelets [33]. The results showed that the tail bleeding time in the ruxolitinib and rhTPO treatment groups was significantly shorter than that of the model group (Figure 6H). These results indicate that the platelets in the ruxolitinib-treated group of mice are fully functional, including the response characteristics to agonists.

### 2.7. Gene Expression Profiling and Functional Analysis of Ruxolitinib-Induced Differentially Expressed Genes

To explore the potential molecular mechanism by which ruxolitinib promotes MK differentiation and platelet production, we performed RNA-seq on ruxolitinib-treated Meg-01 cells to identify gene-wide transcriptomic changes. Based on transcriptome data of 28616 genes, a Venn diagram showed that 12174 DEGs were identified by the RNA-seq method (Figure 7A). Hierarchical clustering analysis revealed systematic changes in gene expression between samples (Figure 7B). To understand the potential role of DEGs regulated by ruxolitinib, GO enrichment analysis was performed on the identified DEGs (Figure 7C). The results showed that the effects of DEGs were mainly involved in the chemokine-mediated signaling pathway, positive regulation of myeloid cell differentiation, response to interleukin-1, regulation of cytokine secretion, positive regulation of cytokine secretion, etc., which are closely related to MK differentiation and platelet production. Furthermore, the signaling pathway regulated by ruxolitinib intervention in Meg-01 cells may be explained by enrichment of the KEGG pathway (Figure 7D). The results showed that ruxolitinib significantly regulated Cytokine-cytokine receptor interactions, the IL-17 signaling pathway, Chemokine signaling pathway, MAPK signaling pathway, Hematopoietic cell lineage, and NF-kappa B signaling pathway, all of which play very important roles in platelet production.

To further explore the molecular mechanisms of ruxolitinib-promoted MK differentiation and platelet production, we selected genes from DEGs that regulate MK development, platelet production and platelet function-related genes (Figure 7E), and we found that, consistent with previous reports, transcription factors such as EGR1, FLI-1, STAT3, RUNX1, FOS, JUNB, ATF4, and MAFF were significantly upregulated after ruxolitinib intervention in MK. In addition, we noted that most of the cytokines that promote MK differentiation and platelet production were significantly upregulated, such as VEGFA, CD14, CD9, CCL5, TLR2, and PPBP; however, the cytokines LDHA, PRMT1, and MYH9 were reported to be negatively regulated in MK differentiation and platelet production, and they were significantly downregulated.

### 2.8. Ruxolitinib Directly Bound to TLR2 to Stimulate MK Differentiation and Platelet Formation

To identify the molecular targets of ruxolitinib in thrombocytopenia, we applied network pharmacology for target prediction analysis. Target prediction was performed by online tools such as PharmMapper and STITCH, and a total of 118 targets were identified (Figure 8A). The PPI results showed that TLR2 could be considered a core protein (Figure 8B). Molecular docking was used to judge the direct relationship between ruxolitinib and its core target TLR2. Docking scores > 5 kcal mol^−1^ were considered high binding intensity [12]. Based on the docking fraction, the binding fraction of TLR2 to ruxolitinib was 6.11, which indicates that ruxolitinib has a good affinity for TLR2 (Figure 8C). To further validate that ruxolitinib promotes MK differentiation, we also showed that TLR2 expression was enhanced with increasing concentrations of ruxolitinib (Figure 8D). A DARTS assay was used to further validate the interaction of ruxolitinib with TLR2. DARTS results showed that ruxolitinib protected against TLR2 degradation caused by protease (Figure 8E,F). Furthermore, the expression of CD41^+^/CD42b^+^, a surface marker for promoting MK differentiation, was detected by loading or not loading the TLR2-specific inhibitor C29. The results showed that combined loading of C29 significantly blocked the effect of ruxolitinib on promoting MK differentiation compared with the Meg-01 group intervened with ruxolitinib alone (Figure 8G,H).

In conclusion, the above results indicate that TLR2 is a potential important target of ruxolitinib to promote MK differentiation and platelet production.

### 2.9. Ruxolitinib Promotes MK Differentiation by Activating Rac1/cdc42/JNK

Randomly selected transcription factors (Figure 9A–D) were verified by immunoblotting, and the results showed that their expression was significantly upregulated by ruxolitinib. All these differential gene changes were associated with ruxolitinib-induced MK differentiation.

Notably, most genes in the Cytokine-cytokine receptor interaction and MAPK signaling pathway were significantly enriched, and both pathways play a very important role in platelet production. Therefore, we speculate that the Rac1/cdc42/JNK pathway is involved in ruxolitinib-induced platelet production. Rac1/cdc42 is reported to regulate the rearrangement of the MK skeleton and can mediate the activation of the MAPK pathway JNK, which in turn promotes MK differentiation and platelet formation [34]. WB results showed rapid and concentration-dependent upregulation of Rac1/cdc42 and P-JNK in the ruxolitinib-treated Meg-01 cell group compared with the control group (Figure 9E,F). The fluorescent expression of the important transcription factor NF-E2 was significantly enhanced (Figure 9G). All these results suggest that ruxolitinib may stimulate MK differentiation, proplatelet formation and platelet release through the cytokine-mediated downstream Rac1/cdc42/JNK pathway.

## 3. Discussion

Thrombocytopenia remains a challenge in clinical hematology and can present serious consequences, such as skin and mucosal bleeding and intracranial and visceral bleeding [35]. Traditional treatment strategies for thrombocytopenia have mainly included platelet transfusion, increasing platelet production, or reducing platelet destruction [36]. Direct platelet transfusion is the most straightforward option to alleviate thrombocytopenia, but it can cause many adverse reactions, such as fever, allergic reactions, and blood-borne diseases [37]. Therefore, increasing the production of platelets is a potential clinical therapy.

Ruxolitinib is a potential candidate drug to promote platelet production and treat thrombocytopenia. Platelet production is considered to be a continuous process that includes MK development and maturation and platelet release [38]. First, K562 and Meg-01 are commonly used models to study MK development in vitro [39]. In the present study, we found that ruxolitinib significantly promoted MK differentiation, as evidenced by increased cell volume, enhanced expression of CD41^+^/CD42b^+^, polyploid formation, and DMS formation. Second, the thrombopoiesis effect of ruxolitinib was evaluated in a mouse model of radiation-induced thrombocytopenia. As expected, we found that ruxolitinib (10 mg/kg) significantly restored the number of circulating platelets in peripheral blood, but it had no therapeutic effect on leukocytes in RI-mice, which indicates that ruxolitinib acts specifically on platelet-forming cells. H&E staining of bone marrow and flow cytometry results showed that ruxolitinib stimulated hematopoietic stem cell differentiation to MKs, MK differentiation, and platelet production. Analysis of H&E staining and flow cytometry results of the extra marrow important hematopoietic organ, the spleen, showed that ruxolitinib treatment also promoted the recovery of splenic MKs. After that, to detect the presence of ineffective platelet production, we performed flow assays on peripheral blood MKs and washed platelets. Consistent with previous results, ruxolitinib significantly restored the number of MKs and platelets in peripheral blood, and platelet function was complete. Overall, these data suggest that ruxolitinib improves thrombocytopenia by restoring and enhancing MK counts and platelet production in the bone marrow, spleen, and peripheral blood.

The whole gene transcriptome data of Meg-01 cells treated with ruxolitinib were analyzed. Through GO enrichment, we found that the positive regulation of myeloid cell differentiation, which is most directly related to platelet formation, was enriched. Most of the DEGs were involved in the chemokine-mediated signaling pathway, response to interleukin-1, regulation of cytokine secretion, positive regulation of cytokine secretion, and other pathways and biological processes, all of which are important in the development of MKs and platelets. The KEGG results also showed significant regulation of Cytokine-cytokine receptor interaction, IL-17 signaling pathway, Chemokine signaling pathway, MAPK signaling pathway, Hematopoietic cell lineage, and NF-kappa B signaling pathway. In addition, the enrichment of differential genes with genes related to MK development, platelet production, and platelet function was consistent with the results of our previous experiments. MK differentiation and platelet production are strictly regulated by cytokines. VEGFA [40], CD14 [41], CD9 [42], CCL5 [43], TLR2 [44], and PPBP [45] have been reported as positive regulatory cytokines for promoting MK differentiation and platelet production. These cytokines were significantly upregulated after ruxolitinib intervention in MK cells. Meanwhile, LDHA [46], PRMT1 [47] and MYH9 [48], which are negative regulators, were significantly downregulated. Transcription factors are important molecules that control gene expression. Analyzing the transcription factors in their differential genes, we found that transcription factors related to MK differentiation and platelet production, such as EGR1 [49], FLI-1 [50], STAT3 [51], RUNX1 [52], FOS [53], JUNB [54], ATF4 [16] and MAFF [55], were also significantly enriched. FOS, JUNB, and ATF4 are all transcripts of AP-1. AP-1 is activator protein-1, which responds to multiple stimuli, including cytokines, growth factors, stress, and bacterial and viral infections, by regulating gene expression [56]. AP-1 controls many cellular processes, including differentiation, proliferation, and apoptosis [57]. Previous studies have reported that AP-1 plays an important role in increasing DNA synthesis in immature MKs [58]. AP-1 increases its transcriptional activity through MAPK signaling and phosphorylation of basal AP-1 by increasing the binding efficiency of the transcriptional activator CBP (CREB binding protein) [59]. In addition, AP-1 activation initiates a positive feedback loop for further AP-1 upregulation [60].

It was previously reported that MK differentiation and platelet production are regulated by TLR signaling [61]. Interestingly, TLR2 was also identified as a potential important target in the network pharmacology prediction study of ruxolitinib for thrombocytopenia. Using molecular docking to study the interaction between ruxolitinib and TLR2, the binding score was 6.11, which also indicated that our binding was stable. The DARTS assay also verified that ruxolitinib significantly increases the stability of TLR2. In addition, loading TLR2 specific inhibitor cannot completely block the MK differentiation promotion effect of ruxolitinib, but the inhibition effect is very significant. This may be due to the fact that TLR2 is the main target of ruxolitinib in promoting MK differentiation, but it is not the only target. Therefore, we speculate that the binding of ruxolitinib to the primary target TLR2 may trigger downstream signaling pathways to promote MK differentiation and platelet formation, which warrants further investigation in further studies.

TLR2 signaling activates multiple downstream pathways, including MAPK and NF-κB [62], which are known to promote MK differentiation and platelet production, while Rac1/cdc42/JNK is an important branch. Rac1 and cdc42, both Rac GTPases in the Rho family of small GTPases, are involved in the regulation of various hematopoietic cell functions, including hematopoietic stem cell proliferation, transplantation, and bone marrow retention, neutrophil chemotaxis and superoxide production, macrophage phagocytosis, and B- and T-cell immune responses [63]. In platelets, activation of Rac1 and cdc42 is associated with actin polymerization, lamellipodia formation, and stability of platelet aggregates under shear stress [64]. Moreover, Rac1 and Cdc42 can regulate actin cytoskeleton remodeling and activate the JNK signaling pathway, thereby promoting cell proliferation and migration [65]. The PI3K/Akt-Rac1-FAK-JNK pathway has been shown to be activated by basic fibroblast growth factor (BFGF), which facilitates melanoma migration [66]. In the current study, we showed that Rac1, cdc42, and JNK are activated downstream of the targeting action of ruxolitinib with TLR2 and are involved in proplatelet formation (Figure 10).

In addition, ruxolitinib is more commonly reported as a selective JAK1/2 inhibitor [67]. However, in the present experiments, we proposed that ruxolitinib targets TLR2 to promote MK differentiation and platelet production. This indicates that drug intervention has different regulatory mechanisms in different biological models due to the complexity of biological models. TLR2, as a supramundane receptor, can activate multiple downstream pathways, but the relationship with JAK has rarely been reported. Moreover, JAK can be activated by multiple upstream receptors. Finally, different drug concentrations and delivery methods also influence the mechanism and effect of drug intervention with unexpected behaviors. For example, 5 μM ruxolitinib inhibits the Stat3 and Akt/mTOR/Yap pathways in TGF-β1-induced NRK-49F cells and thereby relieves renal interstitial fibrosis in UUO mice [68], and ruxolitinib cream was shown to improve skin inflammation and reduce pruritus by downregulating multiple components of the JAK-STAT signaling cascade in studies of fluorescein isothiocyanate (FITC)-induced dermatitis in mice and human skin explants [67]. In our experiments, we found that 5 μM, 10 μM, and 20 μM of ruxolitinib significantly promoted the characteristic differentiation of MKs from K562 and Meg-01 cells, and that 10 mg/mL of ruxolitinib intraperitoneally promoted platelet recovery in radiation-injured mice. In conclusion, these reports all suggest that ruxolitinib has the potential to quantitatively promote MK differentiation and platelet production.

## 4. Materials and Methods

### 4.1. Cell Culture

K562 cells (chronic myelogenous leukemia cells) and Meg-01 cells (human megakaryocytic leukemia cell line) were purchased from American Type Culture Collection (Rockville, MA, USA). The two cell lines were cultured in RPMI 1640 medium (Gibco Life Technologies, Carlsbad, CA, USA) supplemented with 10% fetal bovine serum (FBS, CAT:SP10020500, Sperikon Life Science & Biotechnology Co., Ltd., Chengdu, China) and 1% penicillin-streptomycin solution (Gibco, Invitrogen Corporation, Carlsbad, CA, USA) at 37 °C in a humidified atmosphere with 5% CO_2_.

### 4.2. Cell Proliferation Assay

Cell proliferation of K562 and Meg-01 cells was evaluated by Cell Counting Kit-8 (CCK-8) assay, according to the manufacturer’s instructions (Dojindo, Kyushu, Japan). In brief, K562 cells and Meg-01 cells were seeded in 96-well plates at a density of 5.0 × 10^3^ cells. K562 cells and Meg-01 cells were treated with different concentrations of ruxolitinib (5, 10, and 20 μM) at 37 °C and 5% CO_2_ from day 1 to day 5. The untreated cells were considered the control group. Following the treatment, CCK-8 solution was added to each well and incubated at 37 °C for 2 h. The absorbance (OD) was measured at 450 nm using a microplate reader (BioTek, IL, USA). Each experiment was performed at least in triplicate.

### 4.3. Lactate Dehydrogenase (LDH) Assay

A total of 5.0 × 10^3^ cells was seeded into a 96-well plate and treated for 1, 3, and 5 days with or without ruxolitinib (5, 10 and 20 μM). The cytotoxicity was then determined using an LDH cytotoxicity assay kit (Beyotime, Jiangsu, China) according to the manufacturer’s instructions.

### 4.4. Morphological Observations

Using a safe concentration based on the cell viability assay, after treatment with or without ruxolitinib (5, 10, and 20 μM) and phorbol 12-myristate 13-acetate (PMA, 2.5 nM, Macklin, Shanghai, China) for 5 days in 6-well plates, the cell morphology was observed by light microscopy (Nikon, Japan).

### 4.5. Giemsa Staining

K562 and Meg-01 cells (4.0 × 10^4^) were seeded in 6-well plates and treated with ruxolitinib (5, 10, and 20 μM) and PMA for 5 days. The cells were washed and harvested with phosphate buffered saline (PBS) and swelled with 0.075 M KCl solution. Then, the cells were fixed with fixing solution (methanol: glacial acetic acid = 3:1 (*v*/*v*)), placed onto glass slides and stained with Giemsa solution (Solarbio, Beijing, China) for 8 min. After that, the slides were observed directly, and multinucleated MKs were captured by light microscopy (Nikon, Tokyo, Japan).

### 4.6. Analysis of Cell Differentiation

For the detection of MK differentiation, K562 and Meg-01 cells treated with or without ruxolitinib (5, 10, and 20 μM) and PMA for 5 days were harvested and washed with ice-cold PBS 3 times. Then, 100 μL of the cell suspension at a density of 1.0 × 10^6^ cells/mL was transferred to l mL Eppendorf (EP) tubes followed by incubation with FITC-conjugated anti-CD41 and PE-conjugated anti-CD42b antibodies (Biolegend, San Diego, CA, USA) for 30 min at room temperature in the dark. The cells were analyzed using a BD FACSCanto II flow cytometer (BD Biosciences, San Jose, CA, USA). Each experiment was performed in triplicate.

### 4.7. Phalloidin Staining

Approximately 1×10^6^ K562 and Meg-01 cells were harvested to stain phalloidin and DAPI after drug induction for 5 days. Briefly, cells were resuspended in PBS and harvested onto glass slides through a TD3 cytocentrifuge (Shanghai Lu Xiangyi Centrifuge Instrument Co., Ltd., Shanghai, China). Then, the cells were fixed with 4% paraformaldehyde for 15 min and permeabilized with 0.05% Triton X-100 for 10 min at room temperature. After that, the cells were washed twice with PBS, and TRITC-conjugated phalloidin (1:200) (Solarbio, Beijing, China) was added for 30 min in the dark at room temperature. Then, DAPI (Solarbio, Beijing, China) was added to counterstain the nucleus for 5 min. Finally, representative images were captured using an inverted fluorescence microscope (Nikon Ts2R/FL, Japan) with excitation at 560 nm laser for phalloidin and 405 nm for DAPI.

### 4.8. RIT Establishment in Mice and Ruxolitinib Treatment

Specific pathogen-free (SPF) Kunming (KM) mice, 8 to 10 weeks old and 18–22 g in weight, were purchased from Da-shuo Biotechnology Limited (Chengdu, Sichuan, China). The mice were bred under standard conditions (22 ± 2 °C, 55  ±  5% humidity and 12 h light/dark cycle) and were fed standard diets and allowed to drink freely. All experimental operations were approved by the laboratory animal ethics committee of Southwest Medical University (Luzhou, China, License No. 20220817-008). After acclimation for a week, all mice were randomly assigned to 4 groups: the control group, X-ray (thrombocytopenia model) group, X-ray + rhTPO (3SBIO; Shenyang, China, 1500 U/kg, positive control) group, and X-ray + ruxolitinib (10 mg/kg, treatment) group. In addition to the control group, the other mice were given a single dose of X-ray (4 Gy) to establish a mouse model of thrombocytopenia. Then, the mice in the control group and model group were intraperitoneally administered normal saline per day. The mice in the rhTPO-positive group and ruxolitinib group were intraperitoneally administered rhTPO (1500 U/kg) or ruxolitinib (10 mg/kg) per day for 12 days.

### 4.9. Measurement of Hematologic Parameters

On days 0, 4, 7, 10, and 12, a small amount of peripheral blood (40 μL) was drawn from the eyes’ fundus vein plexus on the indicated days and treated with 160 μL of diluent for hematologic parameter analysis by a hematology analyzer (SYSMEX XT-1800Iv; Kobe, Japan).

### 4.10. Flow Cytometry Analysis of BM, Spleen and Blood Cells

For the analysis of MKs in BM, spleen and blood cells, total BM cells were washed out of the femur with saline solution, and the spleen was ground into single cells and filtered by nylon net. Red blood cells (RBCs) were removed from the cell samples with RBC lysis buffer (Beijing 4 A Biotech, Beijing, China). For analysis of MKs in peripheral blood, 50 μL of blood was taken from the ophthalmic venous plexus and added to an EP tube prefilled with sodium citrate for mixing. The cell density was adjusted to 100×10^4^ per sample by counting on a hematology analyzer. The samples were labeled with FITC-conjugated anti-CD41 (BioLegend, San Diego, CA, USA), PE-conjugated anti-CD117 (c-Kit, BioLegend, San Diego, CA, USA) and FITC-conjugated anti-CD41 (BioLegend, San Diego, CA, USA) and PE-conjugated anti-61 (BD Biosciences, San Jose, CA), FITC-conjugated anti-CD41 and PE-conjugated anti-CD62P (BioLegend, San Diego, CA, USA) on ice for 30 min in the dark. Flow cytometry was performed using a BD FACSCanto II flow cytometer (BD Biosciences, San Jose, CA).

### 4.11. Platelet Isolation and Sample Preparation

Mouse blood was drawn into an anticoagulant tube containing 3.8% sodium citrate (V: V = 1:9) as the anticoagulant. Immediately after collection, blood was centrifuged at 100× *g* for 10 min at room temperature, and the platelet-rich plasma (PRP) was carefully separated. PRP was centrifuged at 400× *g* for 10 min to sediment platelets. The supernatant was pipetted away and discarded, while the platelet pellet was washed and resuspended in modified Tyrode’s buffer (137 mM NaCl, 2.9 mM KCl, 0.34 mM Na_2_HPO_4_, 12 mM NaHCO_3_, 5 mM HEPES, 5 mM glucose, 1 mM MgCl_2_, and 1 mM CaCl_2_, pH 7.3) for the next detections. After that, they were permitted to rest for 1–2 h at 25 °C.

### 4.12. Platelet Activation Analysis

Platelet activation was performed using washed platelets prepared as described above and stimulated with agonist (ADP: 10 μM, Helena Laboratories, USA) for 10 min at 37 °C, after which resting or activated platelets were incubated with PE-conjugated anti-CD62P (BioLegend, San Diego, CA, USA) for 30 min at room temperature in the dark by flow cytometry analysis.

### 4.13. Tail Bleeding Assay

The distal tail of 2 mm of nonanesthetic mice was cut and immediately immersed in normal saline at 37 °C. The bleeding time from transection to initial hemostasis (more than 2 min after stopping) was measured. If the bleeding did not stop, the bleeding time was recorded as 10 min. Be careful not to apply pressure to the tail that may affect homeostasis.

### 4.14. Histology Analysis

After mice were treated with ruxolitinib for 12 days, three mice were randomly selected from each group, and the femur and spleen were separated. The femurs were completely infiltrated in 10% formaldehyde over 24 h and then decalcified with decalcification solution for more than one month. The femurs were then embedded in paraffin, cut into 5 μm sections, stained with hematoxylin and eosin (H&E), and photographed under an Olympus BX51 microscope (Olympus Optical, Japan). Three fields of view of each sample were randomly photographed, and MK was counted.

### 4.15. Acquisition of Candidate Targets of Ruxolitinib against Thrombocytopenia

Structural information of ruxolitinib was obtained from the PubChem database and uploaded in SMILES format to the SwissTargetPrediction database to identify potential targets for ruxolitinib. Additional alternative targets for ruxolitinib were obtained from the GeneCards databases. The GeneCards database and the OMIM database were used to identify targets associated with thrombocytopenia. After removal of duplicates, common targets of ruxolitinib and thrombocytopenia were considered potential targets for ruxolitinib against thrombocytopenia. Finally, Venn diagrams were drawn to obtain overlapping targets.

### 4.16. Protein–Protein Interaction (PPI) Network Construction and Screening of Core Targets

To analyze the target-target interactions, we constructed PPI networks using the STRING database and imported the data into Cytoscape_v3.7.1 software for visibility and topology analysis. Ruxolitinib’s primary anti-thrombocytopenia targets were chosen based on the following criteria: degree values must be greater than or equal to twice the corresponding median, betweenness centrality must be greater than or equal to the corresponding median, and closeness centrality must be greater than or equal to the corresponding median. Degree values larger than 47, betweenness centrality greater than 0.002858932, and closeness centrality greater than 0.507867733 were the screening criteria.

### 4.17. Molecular Docking

Molecular docking is one of the common methods used in structure-based drug design [69]. When small molecule ligands bind to target molecules to form stable complexes, the preferred concept can be predicted, and the binding strength and affinity between them can be calculated. It is widely used to better understand the interaction characteristics of small molecules with therapeutic targets of many pathogens of clinical interest. Here, molecular docking was used to predict the binding affinity between ruxolitinib and the target molecule TLR2. The 3D structure of ruxolitinib was obtained from PubChem. The 3D crystal structures of human TLR2 were obtained from the RCSB Protein Data Bank (https://bivi.co/visualization//rcsb-protein-data-bank) in 18 October 2022. The optimized structure of TLR2 was then obtained by adding hydrogen atoms, adding charge and minimizing energy. The molecular docking between ruxolitinib and TLR2 was simulated using SYBYL-x 2.0 and visualized using PyMOL software. The Surflex-Dock score (total score) indicates the binding affinity.

### 4.18. Drug Affinity Responsive Target Stability Assay (DARTS)

Meg-01 cells were collected, lysed with RIPA lysis buffer on ice for 15 min, and centrifuged at 12,000 rpm/min for 15 min, and the supernatant was obtained. The protein concentration was determined by the Bradford reagent. Before drug treatment, the protein concentration was diluted to 5 mg/mL. The samples were treated with ruxolitinib and DMSO (Sigma-Aldrich, D2650) at room temperature for 1 h. Thereafter, various pronase dilutions (1:500, 1:1000 and 1:1500) were added and incubated at 40 °C for 10 min. Then, 5× loading buffer was added and boiled for 10 min. All parts of each sample were used for western blot analysis. ACTB was used as an internal control.

### 4.19. Cell Sample Preparation

After Meg-01 cells were treated with or without ruxolitinib (20 μM) for 3 days, total RNA from each group was extracted using RNA-easy Isolation Reagent (Vazyme Biotech, Nanjing, China) according to the manufacturer’s instructions. RNA quality was assessed using a 2100 Expert Bioanalyzer (Agilent) and sent for library preparation and sequencing using the Illumina HiSeq2000 platform of Majorbio Biotech (Shanghai, China).

### 4.20. RNA-seq and Data Analysis

RNA-seq data were generated by Shanghai Majorbio Biopharm Biotechnology Co., Ltd. (Shanghai, China). In brief, an RNA-seq transcriptome library was created using 1 mg of total RNA and the TruSeqTM RNA Sample Preparation Kit from Illumina (San Diego, CA, USA). SeqPrep (https://github.com/jstjohn/SeqPrep) and Sickle (https://github.com/najoshi/sickle) were used to trim and quality control the original paired-end readings in July 2022. The clean sequencing reads were aligned to the human genome (hg38) using TopHat software. The data were analyzed using the free Majorbio I-Sanger Cloud Platform (www.i-sanger.com) in 18 October 2022.

### 4.21. Gene Ontology and KEGG Pathway Analysis

The Gene Ontology Consortium created the GO database (http://geneontology.org/) in 18 October 2022, and GO analysis was used to identify the differential expression of genes associated with the primary function as described by the Gene Ontology, illuminating the hierarchical relationship between gene functions in NCBI. Similarly, the KEGG database and differential gene ontology’s major pathways were examined using pathway analysis (https://www.genome.jp/kegg/) in 18 October 2022. GO terms and KEGG pathways with *p*-values < 0.05 were considered significantly enriched by differentially expressed genes (DEGs).

### 4.22. Western Blotting

Meg-01 cells were harvested on the fifth day after treatment with ruxolitinib (5, 10 and 20 μM). Total protein was extracted from cells that had undergone various treatments using 1 × RIPA lysis buffer (CST, MA, USA) in addition to protease inhibitors (Sigma, St. Louis, MO, USA) and phosphatase inhibitors (Roche, Penzberg, Germany). The protein was quantified using the Quick StartTM Bradford 1 × Dye Protein Assay Reagent (Bio-Rad, Hercules, CA, USA). Equal amounts of protein (25 μg) were electrophoretically separated using 7.5%, 10% sodium dodecyl sulfate-polyacrylamide gel electrophoresis (SDS-PAGE) and then transferred to polyvinylidene fluoride (PVDF) membranes after being heated at 95 °C for 10 min. The membrane was blocked in 5% skimmed milk powder for 60 min, followed by incubation with primary antibodies overnight at 4 °C. Then, HRP-labeled secondary antibodies were added and incubated for 1 h at room temperature after being washed (three times) with PBS with Tween 20 (PBST). The ChemiDoc MP Imaging System was used to detect the protein bands after the protein bands were seen using the ECL Western Blotting detection reagent (4A Biotech Co., Ltd., Beijing, China) (Bio-Rad, Hercules, CA, USA), and the gray value of the protein bands was quantified with ImageJ software. The relative image intensity of the target protein and GAPDH positively shows their expression. The primary antibodies were as follows: FOS (Proteintech, USA, 66590-1-lg), EGR1 (Proteintech, USA, 22008–1-AP), RUNX1 (Proteintech, USA, 25315-1-AP), TLR2 (Proteintech, USA, 17236-1-AP), Rac1/cdc42 (CST, USA, 4651S), JNK (Abmart, China, T55490), p-JNK (Abmart, China, T55541), NF-E2 (Proteintech, USA, 11089-1-AP), and GAPDH (Proteintech, USA, 60004-1-lg).

### 4.23. Immunofluorescence Assay

After treatment with ruxolitinib for 5 days, K562 and Meg-01 cells were harvested and resuspended in PBS and harvested onto slides via a D3 cell centrifuge (Shanghai Lu Xiangyi Centrifuge Instruments Co., Ltd., Shanghai, China). Cells were fixed with 4% paraformaldehyde for 15 min and permeabilized with 0.05% Triton X-100 for 10 min at room temperature. After that, the cells were washed twice with PBS and blocked with 5% BSA for 60 min. Then, primary antibodies against NF-E2 (1:100, Proteintech, 11089-1-AP) were incubated with the cells overnight at 4 °C. The cells were then incubated with secondary FITC-labeled goat anti-rabbit antibody (1:200; Zhongshan Golden Bridge Biotechnology, Beijing, China) for 1 h at room temperature, rinsed with PBS, counterstained with DAPI, and analyzed under a fluorescence microscope with excitation wavelengths at 470 nm for FITC and 405 nm for DAPI.

### 4.24. Statistical Analysis

The data collected in this study were statistically analyzed with GraphPad Prism 8.0 (GraphPad Software Inc., La Jolla, CA, USA). In this study, all results are presented as the mean standard deviations (SDs) of at least three independent experiments. Two-tailed Student’s t tests and one-way analyses of variance were used to determine the statistical significance (ANOVA). A *p*-value of 0.05 or lower was regarded in every instance as proving a meaningful difference.

## 5. Conclusions

In our study, the pharmacological action and molecular mechanism of ruxolitinib in RIT were elucidated for the first time. Our study proves that ruxolitinib can significantly promote MK differentiation in vitro. In vivo, ruxolitinib restores MK recovery in the bone marrow and spleen and significantly increases peripheral platelet counts. Further studies proved that ruxolitinib can bind directly to TLR2 to activate the Rac1/cdc42/JNK pathway, leading to differentiation of MKs and anti-RIT effects. In conclusion, these data provide promising information to elucidate the material basis and mechanism of ruxolitinib for RIT treatment and report the first assessment of the role of ruxolitinib in hematopoiesis and as a radiation palliative agent.

## Figures and Tables

**Figure 1 ijms-23-16137-f001:**
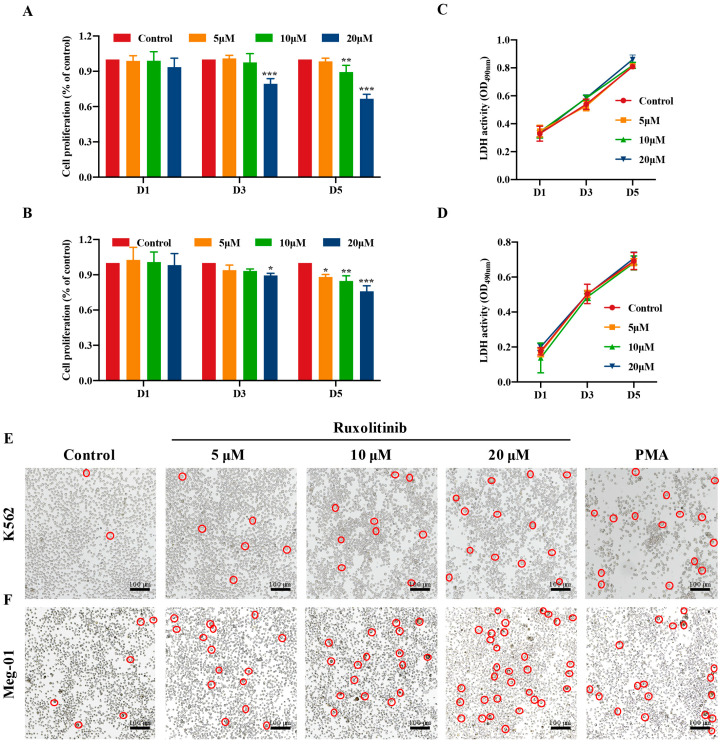
Safe concentration of ruxolitinib for treatment of K562 and Meg-01. (**A**, **B**) The effect of Ruxolitinib intervened on MKs proliferation. Different time points and concentrations on the proliferation rate (%) of megaryocytes. Results of the CCK-8 assay for K562 and Meg-01 cells proliferation; (**C**, **D**) The LDH release of ruxolitinib intervened K562 and Meg01 cells at different time points; (**E**, **F**) Representative images of K562 and Meg-01 cells treated with various concentrations of Ruxolitinib (5, 10, and 20 μM) for 5 days. The positive control is PMA (2.5 nM). n = 3, * *p* < 0.05, ** *p* < 0.01, *** *p* < 0.001, vs. control.

**Figure 2 ijms-23-16137-f002:**
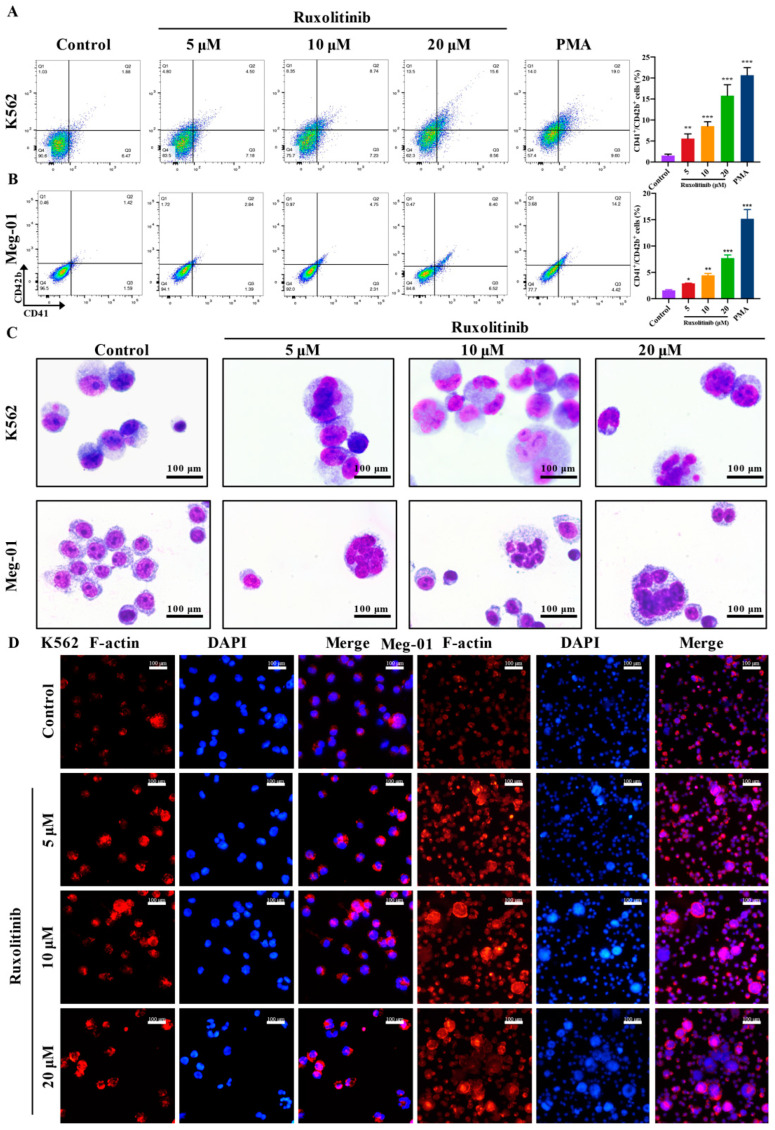
Ruxolitinib induces dramatic morphological changes and differentiations. (**A**,**B**) The percentage of CD41^+^/CD42b^+^ complexes surface expression on K562 and Meg-01 cells by ruxolitinib (5, 10 and 20 μM) or PMA (2.5 nM) for 5 days and analyzed by flow cytometry; (**C**) Giemsa staining of K562 and Meg-01 cells treated with ruxolitinib (5, 10, and 20 μM). Magnification: 400×, Scale bar: 100 μm; (**D**) Phalloidin-labeled cytospin in K562 and Meg-01 cells on day 5 under a fluorescence Microscope (excitation wavelength: 560 nm for Phalloidin, 405 nm for DAPI). Magnification: 400×, Scale bar: 50 μm. n = 3, mean ± SD. Statistics were determined by one-way ANOVA with Dunnett’s and two-way ANOVA with Tukey’s 862 multiple comparisons test, * *p* ˂ 0.05, ** *p* ˂ 0.01, *** *p* ˂ 0.001 vs. the control.

**Figure 3 ijms-23-16137-f003:**
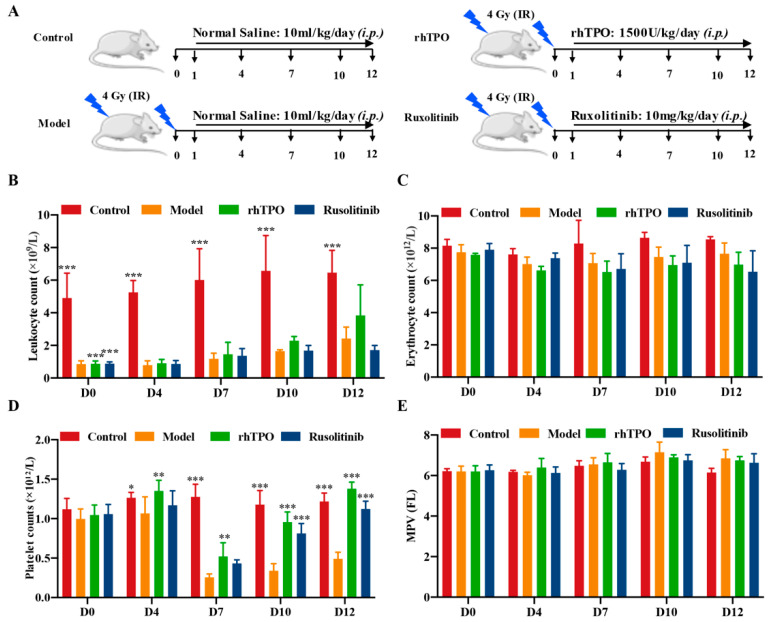
Ruxolitinib mitigates IR-induced Thrombocytopenia in mice. (**A**) Radiation and dosing strategies in mice; (**B**–**E**) Blood counts showing (**B**) WBC, (**C**) RBC, (**D**) platelet counts and MPV on days 1, 4, 7, 10, 12 post-IR. (n = 6 per group) The data are expressed as the mean ± SD. Two-way ANOVA with Tukey’s multiple comparisons test was used unless otherwise specified, * *p* ˂ 0.05, ** *p* ˂ 0.01, and *** *p* ˂ 0.001, vs. Model.

**Figure 4 ijms-23-16137-f004:**
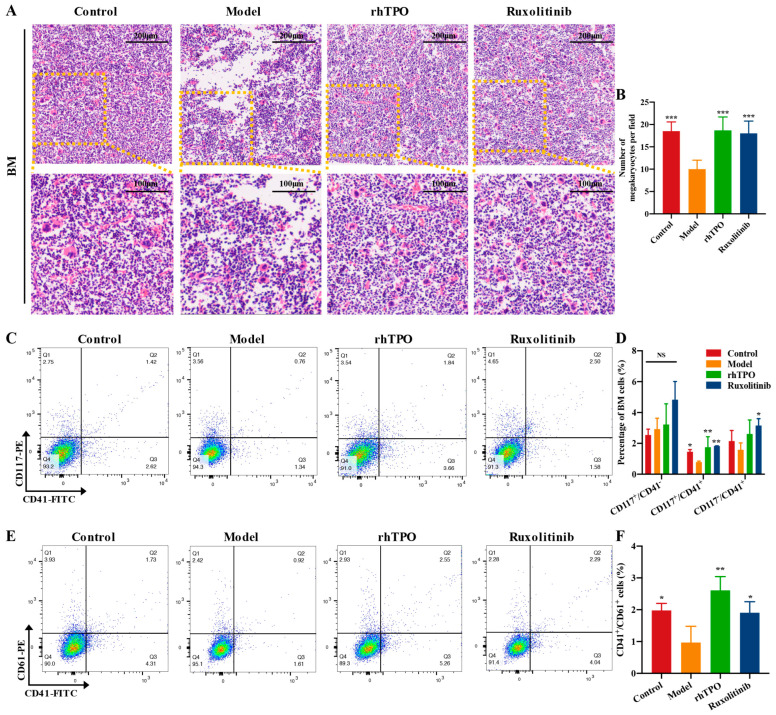
Ruxolitinib rescues bone marrow MKs post radiation injury. (**A**) Images of H&E staining of BM taken with a microscope at magnifications of 100× (top) and 200× (bottom); (**B**) The number of MKs in each group is indicated by the histogram. The data represent the mean standard deviation of three independent experiments; (**C**) The examination of the expression of c-Kit and CD41 in each group by flow cytometry after receiving therapy for 12 days; (**D**) The histogram represents the percentage of c-Kit^+^CD41^−^, c-Kit^+^CD41^+^, and c-Kit^−^CD41^+^ cells in each groups; (**E**) The examination of the expression of CD41 and CD61 in each group by flow cytometry after receiving therapy for 12 days; (**F**) The histogram represents the percentage of CD41^+^CD61^+^ cells in each group. Data represent the mean ± SD of three independent experiments. * *p* < 0.05, ** *p* < 0.01, *** *p* < 0.001 vs. the model group.

**Figure 5 ijms-23-16137-f005:**
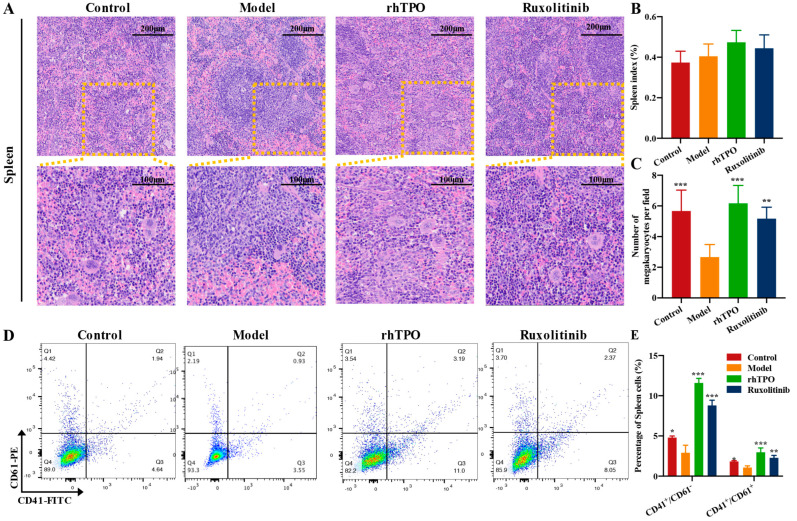
Ruxolitinib restores splenic hematopoiesis. (**A**) Images of H&E staining of spleen taken with a microscope at magnifications of 100× (top) and 200× (bottom); (**B**) Spleen-body weight ratio (n = 6 per group); (**C**) The number of MKs in each group is indicated by the histogram; (**D**) The examination of the expression of CD41 and CD61 in each group by flow cytometry after receiving therapy for 12 days; (**E**) The histogram represents the percentage of CD41^+^CD61^+^ cells in each groups. Data represent the mean ± SD of three independent experiments. * *p* < 0.05, ** *p* < 0.01, *** *p* < 0.001 vs. the model group.

**Figure 6 ijms-23-16137-f006:**
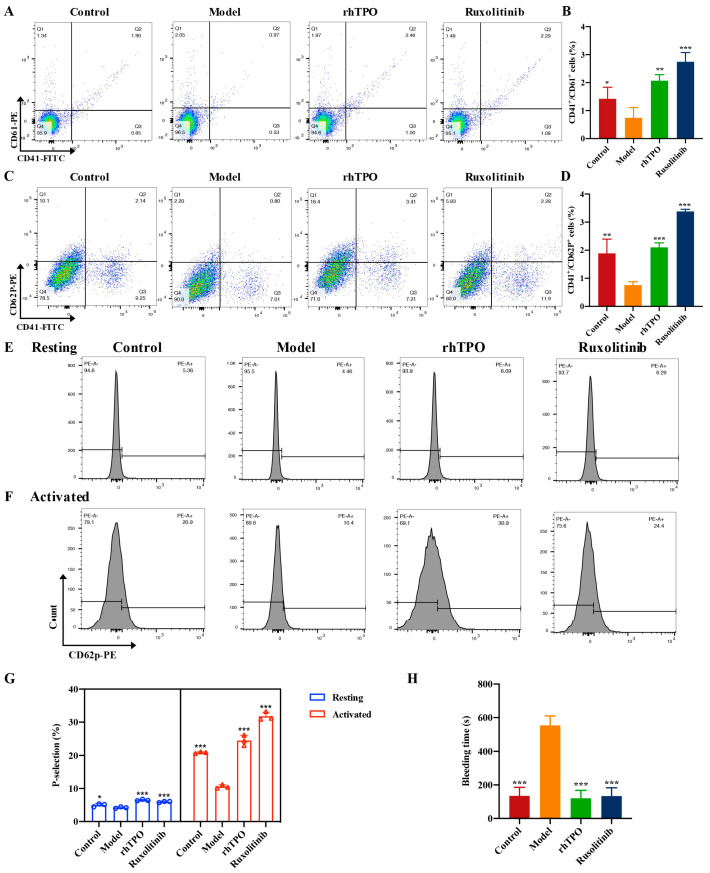
Ruxolitinib promotes the number and function recovery of peripheral blood platelets. (**A**) The examination of the expression of CD41 and CD61 in each group by flow cytometry after receiving therapy for 12 days; (**B**) The number of CD41^+^/CD61^+^ MKs in each group is indicated by the histogram; (**C**) The examination of the expression of CD41/CD62P in each group by flow cytometry after receiving therapy for 12 days; (**D**) The number of CD41^+^/CD62P^+^ MKs in each group is indicated by the histogram; (**E**) Representative images of CD62p in washed platelets in Control, Model, rhTPO, and ruxolitinib groups; (**F**) Representative images of CD62p in ADP (10 µM) washed platelets in Control, Model, rhTPO, and ruxolitinib groups; (**G**) Statistical analysis results of CD62p ratio in washed platelets with or without ADP (10 µM) loading in Control, Model, rhTPO, and ruxolitinib groups; (**H**) Statistical chart of tail clotting time of mice in Control, Model, rhTPO, and ruxolitinib groups. The data represent the mean standard deviation of three independent experiments. * *p* < 0.05, ** *p* < 0.01, *** *p* < 0.001 vs. the model group.

**Figure 7 ijms-23-16137-f007:**
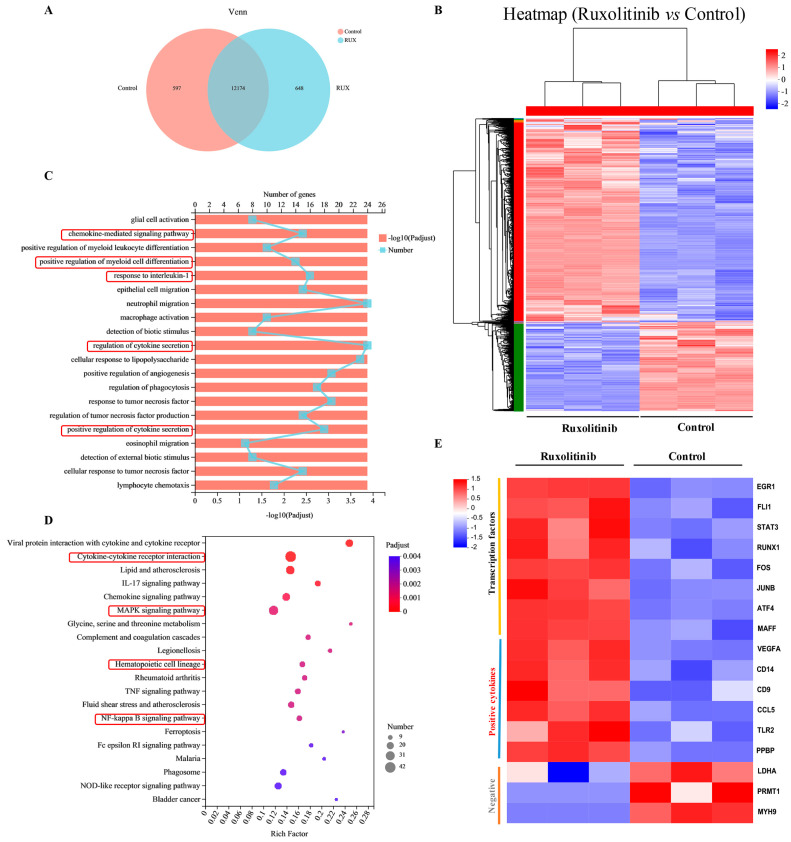
Bioinformatics analysis of gene expression profiles. (**A**) Venn diagram of the identified DEGs; (**B**) Hierarchical clustering analysis of DEGs regulated by ruxolitinib; (**C**) GO enrichment analysis of the identified DEGs; (**D**) KEGG pathway enrichment analysis of the identified DEGs; (**E**) Cluster analysis of DEGs associated with ruxolitinib regulated MK differentiation and platelet production.

**Figure 8 ijms-23-16137-f008:**
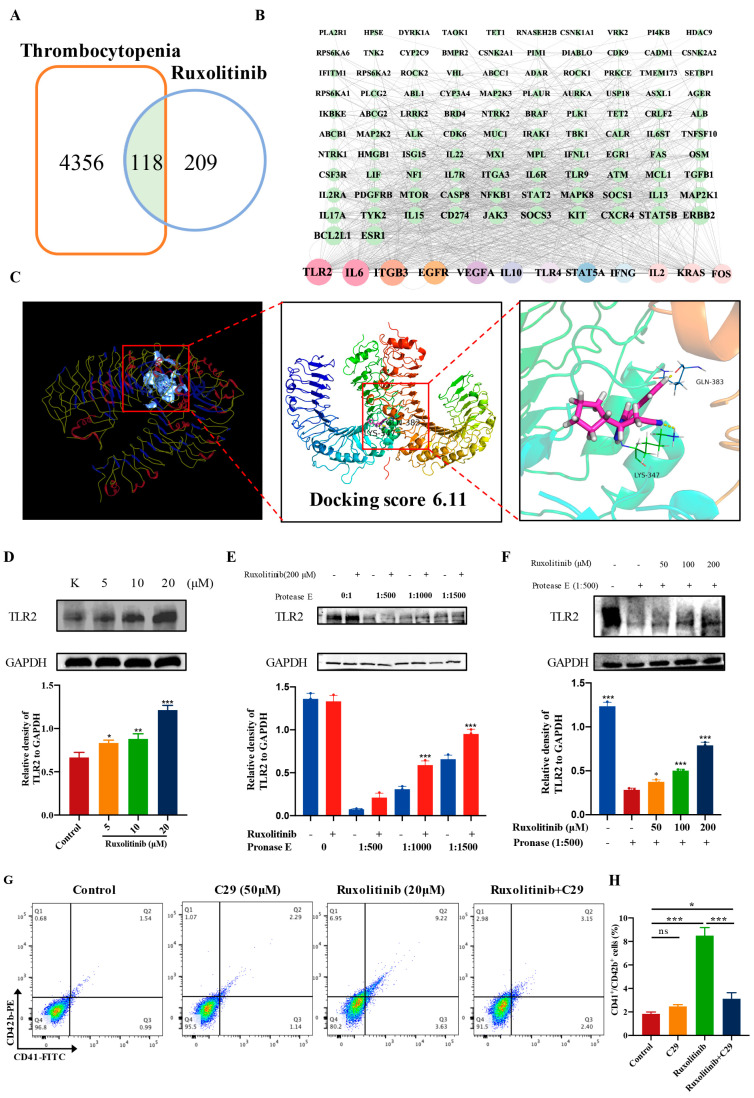
Biophysical validation reveals TLR2 as a new protein target of ruxolitinib. (**A**) Venn diagram shows the common targets of Ruxolitinib and thrombocytopenia. The intersecting part represents the common targets between Ruxolitinib and thrombocytopenia; (**B**) PPI network for identifying core targets of Ruxolitinib against thrombocytopenia through the screening conditions of Degree > 47, BC > 0.002858932, CC > 0.507867733; (**C**) TLR2 and ligands (ruxolitinib) by molecular docking; (**D**). Representative immunoblot images and biochemical quantification of TLR2 after treatment with Ruxolitinib (5, 10, and 20 μM) in Meg-01 cells for 5 day (**E**) The DARTS assay for target validation. TLR2 protein stability was increased upon Ruxolitinib (200 μM) treatment in Meg-01 lysates. Pronase was added using several dilutions (1:500, 1:1000, or 1500) from 50 μg/mL stock for 10 min at 40 °C; (**F**) The DARTS assay demonstrated the dose-dependent binding of Ruxolitinib to TLR2. Treatment with pronase (1:1000) was conducted for 10 min at 40 °C; (**G**) Meg-01 cells were treated with ruxolitinib (20 μM), C29 (50 μM), ruxolitinib (20 μM) + C29 (50 μM) for 5 days. FCM analysis of the expression of CD41 and CD42b. (**H**) The histogram shows the percentage of CD41^+^/CD42b^+^ cells for each group. n = 3, * *p* ˂ 0.05, ** *p* ˂ 0.01, and *** *p* ˂ 0.001. ns: no significance.

**Figure 9 ijms-23-16137-f009:**
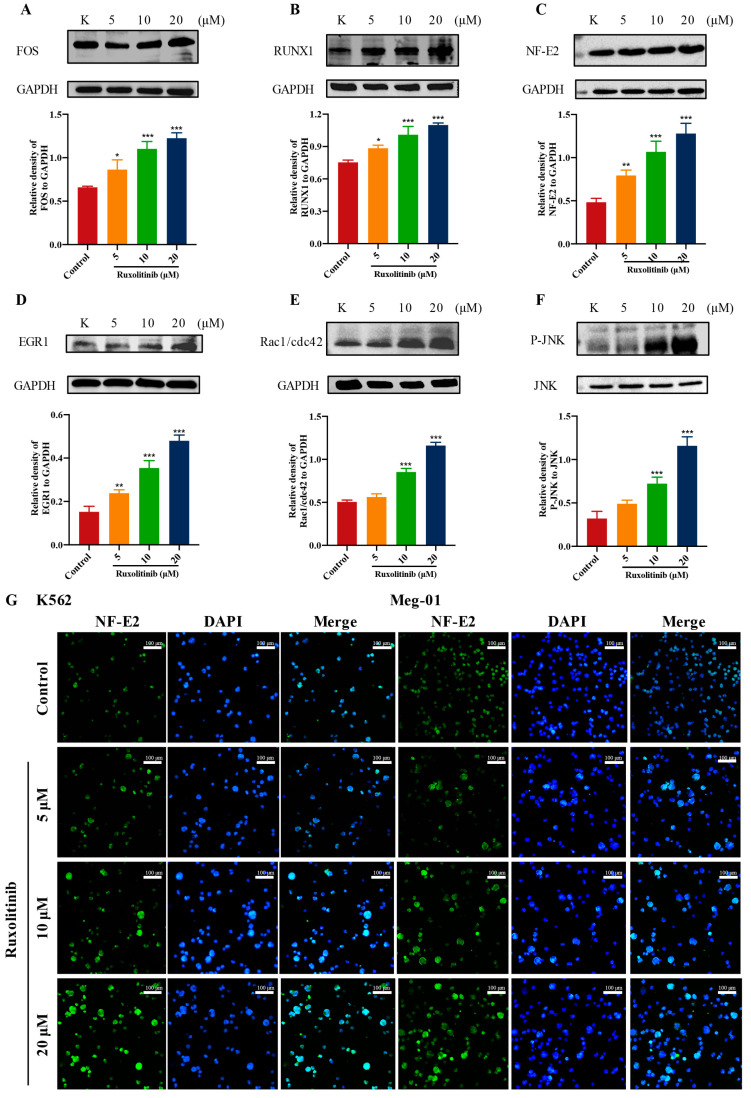
Expression analysis and regulate relationship of DEGs related to MK differentiation. (**A**–**D**) WB analysis of transcription factors related to MK differentiation in selected DEGs; (**E**,**F**) Representative immunoblot images and biochemical quantification of MKs-affiliated pathway proteins (Rac1/cdc42/JNK pathways) after treatment with ruxolitinib (5, 10, and 20 μM) in Meg-01 cells for 5 days; (**G**) Representative immunofluorescence image of the nuclear translocation of NF-E2 in K562 and Meg-01 cells upon treatment with ruxolitinib for 5 days. 470 nm for FITC and 405 nm for DAPI, Magnification: 200×, Scale bar: 100 μm). n = 3, mean ± SD. Statistics were determined by one-way ANOVA with Dunnett’s test, * *p* ˂ 0.05, ** *p* ˂ 0.01, and *** *p* ˂ 0.001 vs. the control.

**Figure 10 ijms-23-16137-f010:**
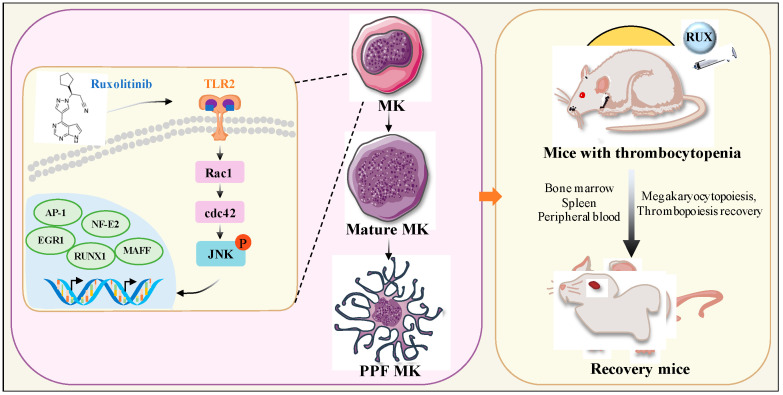
Schematic illustration of the role of ruxolitinib in MK differentiation and platelet pro-duction. Ruxolitinib induces the expression of various cytokines and TLR2, activates the Rac1/cdc42/JNK signaling pathway, and leads to the expression of AP-1, EGR1, RUNX1, and NF-E2. As a result, the activation of AP-1, EGR1, RUNX1, NF-E2 promote the expression of genes related to MK differentiation and thrombopoiesis. These genes contribute to MK maturation and platelet formation and promote the recovery of bone marrow and spleen MKs and accelerate platelet production in RI-mice. PPF: proplatelet-forming MK.

## Data Availability

All figures and data used to support this study are included within this article.

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
