# Peer review of "Targeting TLR2/Rac1/cdc42/JNK Pathway to Reveal That Ruxolitinib Promotes Thrombocytopoiesis"

_ijms, 2022, doi:10.3390/ijms232416137_

Round 1

Reviewer 1 Report

1、 Authors are suggested to check that reference are cited correctly. For example, reticulocytes described in the main text are not related to REF 3;

2、 Reference 13 does not involve JNK description of the role of megakaryocyte and platelet, please change to other correct literature;

3、 Rac1 and Cdc42 were not covered in Reference 60, please cite literatures properly.

4、 Proper referencing should be added throughout the manuscript for better presentation, e.g.:1Additionally, it has been noted that platelet count, more so than any other hematologic marker, is more closely related to survival rate following total body irradiation. line47-49);(2In discussion, It was previously reported that megakaryocyte differentiation and platelet production are regulated by TLR signaling (line :433-434), it is suggested that literature support be added after this.3 Previous studies have reported that AP-1 plays an important role in increasing DNA synthesis in immature MKs This sentence can be found in this article: doi: 10.1074/jbc. M115.707174 (line 433-434).

5、 Some of the experimental results were not clearly expressed: in Results 2.1, “In contrast to the cells receiving culture media, the proliferation of K562 and Meg-01 cells was greatly inhibited (Figures 1-A, B)”. Based on the figures, the inhibition did not occur at all concentrations and time intervals?

6、 In Fig 1E, cell pictures were shown without Scale bar or magnifications; In Figure 4 legend,  “Ruxolitinib rescues bone marrow megakaryocytes post radiation injury. (A) Images of H&E staining of BM taken with a microscope at magnifications of 200× (top) and 100× (bottom)”, the description should be 100× (top) and 200× (bottom) instead; Also in Figure 4A, the enlarged images of TPO and Ruxolitini are not originated from the full-view sources.

7、 In Results 2.8, the concentration of Ruxolitinib up to 20 uM against TLR was used. Why only 10 uM of Ruxolitinib was selected in FIG. 8G? Is the inhibitory effect of C29 on 20um of Ruxolitinib not obvious? Or any other reasons? Moreover, TLR2, a target of Ruxolitinib, is also an important target in this study. In order to make the experimental structure more reliable, it is recommended to use Si-RNA of TLR2 for validation.

8、 In Results 2.7, “WB results showed rapid and concentration-dependent upregulation of Rac1/cdc42 and P-JNK in the ruxolitinib-treated Meg-01 cell group compared with the control group (Figure 9-G, H)”, should be “Figures 9E,F” instead of “Figure 9-G, H”; “The fluorescent expression of the important transcription factor NF-E2 was significantly enhanced (Figure 9-I)” , should be “Figure 9-G” instead of “Figure 9-I”.

9、 Fig 9-A, “Fos and the WB results that provide the antibody show inconsistency”, please indicate the location of Fos in figure; GAPDH in Fig 9B should be aligned with RUNX1; Fig9-D, the resolution of image for EGR1 is in poor quality, please provide a high resolution picture.

10、 In the disscusion 400 line, "but its toxicity to leukocytes was not reduced" . Do authors want to express ruxolitinib has no therapeutic effect on leukocytes in IR-mice? If yes, please revise the sentence for better presentation. If no, please provide more evidence to support.

11、 Fig10: In the Schematic illustration (left), the relationship or possible association between ruxolitinib, rac1, cd42, P-JNK and intranuclear transcription factors and AP-1 is not shown; Any relationship between MK and mature MK? Please also clarify the abbreviation “PPF MK”. It is suggested to revise this figure for better presentation.

Author Response

Dec. 13, 2022

Dear Expert Reviewer,

Thank you very much for the prompt review process and excellent comments. We greatly appreciate the time and efforts which you have spent on it. We are submitting the revised manuscript entitled “Targeting TLR2/Rac1/cdc42/JNK pathway to reveal that ruxolitinib promotes thrombocytopoiesis” (ID: ijms-2075065) to International Journal of Molecular Sciences.

We have carefully considered your comments and suggestions, and addressed each of the concerns in response to the comments (see point by point response). We have revised the manuscripts based on your comments and carefully checked throughout the manuscript and corrected the language errors. Our point-by-point responses to the comments (in blue) are shown below (in red).

1、 Authors are suggested to check that reference are cited correctly. For example, reticulocytes described in the main text are not related to REF 3;

Response: Thank you for your careful reading. We have cited the correct references (page 2, line 47).

2、 Reference 13 does not involve JNK description of the role of megakaryocyte and platelet, please change to other correct literature;

Response: Thanks a lot for your reminding. The reference 13 was mislabeled, we added the correct reference (page 2, line 74).

3、 Rac1 and Cdc42 were not covered in Reference 60, please cite literatures properly.

Response: We have cited the correct reference (page 15, line 490).

4、 Proper referencing should be added throughout the manuscript for better presentation, e.g.:(1)Additionally, it has been noted that platelet count, more so than any other hematologic marker, is more closely related to survival rate following total body irradiation. (line:47-49);(2)In discussion, It was previously reported that megakaryocyte differentiation and platelet production are regulated by TLR signaling (line :433-434), it is suggested that literature support be added after this.(3)  Previous studies have reported that AP-1 plays an important role in increasing DNA synthesis in immature MKs This sentence can be found in this article: doi: 10.1074/jbc. M115.707174 (line :433-434).

Response: Thank you very much for your carefully reading and kindly remindering. We carefully checked the whole manuscript and cited the correct references in the revised manuscript (page 2, line 49; page 15, line 469, 463).

5、 Some of the experimental results were not clearly expressed: in Results 2.1, “In contrast to the cells receiving culture media, the proliferation of K562 and Meg-01 cells was greatly inhibited (Figures 1-A, B)”. Based on the figures, the inhibition did not occur at all concentrations and time intervals?

Response: Thank you for your critical thinking. We are sorry that we didn't express our meaning clearly. Figure 1A and B showed the proliferation rate of K562 and Meg-01 cells treated with or without ruxolitinib at different time points and different concentrations. Control group is the cells receiving culture media. In contrast to control group, the proliferation of K562 and Meg-01 cells was inhibited by ruxolitinib treatment (Figures 1-A, B). We could find that at day 3, K562 and Meg-01 treated with 20 μM ruxolitinib exhibited a decrease in cell proliferation rates. And at day 5, Meg-01 cells exhibited inhibition of proliferation at only 5 μM, and K562 cells exhibited significant inhibition of proliferation at 10 μM. We have rewritten this section (page 3, line 110-114).

6、 In Fig 1E, cell pictures were shown without Scale bar or magnifications; In Figure 4 legend,  “Ruxolitinib rescues bone marrow megakaryocytes post radiation injury. (A) Images of H&E staining of BM taken with a microscope at magnifications of 200× (top) and 100× (bottom)”, the description should be 100× (top) and 200× (bottom) instead; Also in Figure 4A, the enlarged images of TPO and Ruxolitini are not originated from the full-view sources.

Response: Thanks a lot for the excellent suggestion. According to your suggestions, we have added the scale bar in new Figure 1E and magnification of the H&E staining images in Figure 4 and 5 legends (page 7, line 226; page 8, line 252).

In Figure 4, we are sorry for confusing the enlarged picture of H&E staining of BM in the TPO and ruxolitinib groups due to our negligence. We have replaced it with the correct enlarged pictures (New Figure 4-A).

7、 In Results 2.8, the concentration of Ruxolitinib up to 20 uM against TLR was used. Why only 10 uM of Ruxolitinib was selected in FIG. 8G? Is the inhibitory effect of C29 on 20um of Ruxolitinib not obvious? Or any other reasons? Moreover, TLR2, a target of Ruxolitinib, is also an important target in this study. In order to make the experimental structure more reliable, it is recommended to use Si-RNA of TLR2 for validation.

Response: Thank you for your rigorous thinking. As you suggested, we used 20 μM ruxolitinib combined with C29 to interfere with Meg-01 cells. The results showed that the megakaryocyte differentiation induced by ruxolitinib (20 μM) was significantly suppressed by C29 (new Figure 9-G, H). The expression of CD41 and CD42b in ruxolitinib (20 μM) and C29 co-treated group was obvious lower than that of ruxolitinib (20 μM)-treated group (new Figure 9-G, H), indicating that TLR2 mediated ruxolitinib-induced megakaryocyte differentiation. But the megakaryocyte differentiation promoting effect of 20 μM ruxolitinib was not completely blocked to the normal level by C29 (new Figure-H), which suggested that there might be other targets or signaling pathways medicated megakaryocyte differentiation induced by ruxolitinib. We have taken a discussion in the discussion section (page 15, line 473-479).

siRNA silencing is a good way to verify the function of a target. However, at present, nucleic acid transfection of suspension cells is a hot problem not only the positive rate of cell transfection is very limited, but also it is difficult to achieve the ideal knockdown effect [1]. Due to the limited time and condition, we were unable to complete this experiment. Here, we used a suitable alternative approach to demonstrate TLR2 as a target of ruxolitinib for megakaryocyte differentiation. C29, a specific inhibitor of TLR2 [2], was used. We used a combination of ruxolitinib and C29 administration to investigate the target of ruxolitinib for megakaryocyte differentiation. In addition, DARTS assay was also applied. The DARTS method is capable of revealing drug-target interactions from cells or tissues by tracking changes in the stability of proteins acting as receptors of bioactive SMs [3]. Therefore, from the above experiments, we can conclude that TLR2 is an important target of ruxolitinib in promoting megakaryocyte differentiation. Many thanks to your valuable suggestions, which will be of great reference value for our subsequent studies.

Reference:

  1. Ensenauer, R.; Hartl, D.; Vockley, J.; Roscher, A.A.; Fuchs, U. Efficient and gentle siRNA delivery by magnetofection. Biotech Histochem 2011, 86, 226-231, doi:10.3109/10520291003675485.
  2. Zhang, W.; Zhou, H.; Jiang, Y.; He, J.; Yao, Y.; Wang, J.; Liu, X.; Leptihn, S.; Hua, X.; Yu, Y. Acinetobacter baumannii Outer Membrane Protein A Induces Pulmonary Epithelial Barrier Dysfunction and Bacterial Translocation Through The TLR2/IQGAP1 Axis. Front Immunol 2022, 13, 927955, doi:10.3389/fimmu.2022.927955.
  3. Lomenick, B.; Hao, R.; Jonai, N.; Chin, R.M.; Aghajan, M.; Warburton, S.; Wang, J.; Wu, R.P.; Gomez, F.; Loo, J.A.; et al. Target identification using drug affinity responsive target stability (DARTS). Proc Natl Acad Sci U S A 2009, 106, 21984-21989, doi:10.1073/pnas.0910040106.

8、 In Results 2.7, “WB results showed rapid and concentration-dependent upregulation of Rac1/cdc42 and P-JNK in the ruxolitinib-treated Meg-01 cell group compared with the control group (Figure 9-G, H)”, should be “Figures 9E,F” instead of “Figure 9-G, H”; “The fluorescent expression of the important transcription factor NF-E2 was significantly enhanced (Figure 9-I)” , should be “Figure 9-G” instead of “Figure 9-I”.

Response: Thank you for your careful reading and kind reminding. We are sorry that we did not express the order of results correctly, and in the revised version, we have modified it (page 13, line 388, 390).

9、 Fig 9-A, “Fos and the WB results that provide the antibody show inconsistency”, please indicate the location of Fos in figure; GAPDH in Fig 9B should be aligned with RUNX1; Fig9-D, the resolution of image for EGR1 is in poor quality, please provide a high resolution picture.

Response: Thanks a lot for the constructive and careful suggestion. In Figure 9-A, we are very sorry for not correctly identifying the sites of the FOS protein bands. According to the instructions of Proteintech (FOS: Proteintech, USA, 66590-1-lg), the calculated molecular weight is 41KD, but the observed molecular weight is 55-60KD, so we have exposed in a larger range. The results showed that in the original data, the first band at the top is the target band, and the second band at the bottom is the heteroband. In the revised version, we have corrected it (new Figure 9-A).

In Figure 9-B, we have adjusted the visualization area of GAPDH so that GAPDH is aligned with RUNX1.

In Figure 9-D, we are very sorry that there are low-quality data presentation, in the revised manuscript, we have uploaded the image to make the results clearer.

10、 In the disscusion (400 line), "but its toxicity to leukocytes was not reduced" . Do authors want to express ruxolitinib has no therapeutic effect on leukocytes in IR-mice? If yes, please revise the sentence for better presentation. If no, please provide more evidence to support.

Response: We are sorry for our indistinct expression. In revised manuscript, we drew a better description about this point (page 14, line 423).

11、 Fig10: In the Schematic illustration (left), the relationship or possible association between ruxolitinib, rac1, cd42, P-JNK and intranuclear transcription factors and AP-1 is not shown; Any relationship between MK and mature MK? Please also clarify the abbreviation “PPF MK”. It is suggested to revise this figure for better presentation.

Response: Thank you for your rigorous thinking and kind reminder. First, figure 10 is a schematic diagram of the role of ruxolitinib in MK differentiation and platelet production. Ruxolitinib induces the expression of various cytokines and TLR2, activates the Rac1/cdc42/JNK signaling pathway, and leads to the expression of AP-1, EGR1, RUNX1 and NF-E2. As a result, the activation of AP-1, EGR1, RUNX1, NF-E2 promote the expression of genes related to MK differentiation and thrombopoiesis. These genes contribute to MK maturation and platelet formation, and promote the recovery of bone marrow and spleen MKs and accelerate platelet production in RI-mice (page 16, line 521-527). Second, MKs originate from hemopoietic stem cells in bone marrow, and then go through an endomitotic process. During this process, DNA is continuously copied, but the cytoplasm does not divide, thus forming mature MKs [4]. Finally, I'm sorry that we didn't clarify the meaning of PPF MK. PPF MK is called proplatelet-forming megakaryocyte. MKs are highly specialized precursor cells that produce platelets through the process of PPF, which is also described as cytoplasmic extension [5](page 16, line 527). In the revised version, we have added and modified it accordingly.

Reference:

  1. Moreau, T.; Evans, A.L.; Vasquez, L.; Tijssen, M.R.; Yan, Y.; Trotter, M.W.; Howard, D.; Colzani, M.; Arumugam, M.; Wu, W.H.; et al. Large-scale production of megakaryocytes from human pluripotent stem cells by chemically defined forward programming. Nat Commun 2016, 7, 11208, doi:10.1038/ncomms11208.
  2. Pan, J.; Lordier, L.; Meyran, D.; Rameau, P.; Lecluse, Y.; Kitchen-Goosen, S.; Badirou, I.; Mokrani, H.; Narumiya, S.; Alberts, A.S.; et al. The formin DIAPH1 (mDia1) regulates megakaryocyte proplatelet formation by remodeling the actin and microtubule cytoskeletons. Blood 2014, 124, 3967-3977, doi:10.1182/blood-2013-12-544924.

Thank you for all the valuable and helpful comments and suggestions. 

Best regards,

Jianming Wu

Reviewer 2 Report

Ruxolitinib is used as a selective JAK1/2 inhibitor with moderate tyrosine kinase inhibitory activity (Tyk2). The present study demonstrated that Ruxolitinib could act as a TLR2 agonist to activate  Rac1/cdc42/JNK pathway, leading to stimulate megakaryocyte differentiation and accelerate recovery of megakaryocytes and thrombocytopoiesis in a mouse model of radiation-injured thrombocytopenia (RIT).  There are some concerns as listed in the following:

(1) To consider the drug target specificity and the side effect, the authors should give any comments on the effective concentration for TLR2 activation vs. JAK1/2 inhibition as well as the relationship between TLR2-mediated signaling pathway and JAK1/2.

(2) The result statements of 2.7 did not match the Figure 9.

(3) Typos and others:

L31: TLR2 full name first

L73: differentiation survival and apoptosis

L126: * p <.05

L162: TPO full name first

*L176: change the sequence of TPO label and Model label in Figure 3B

*L214, L239: at magnifications of 200× (top) and 100× (bottom)?

L272: Figure 6G

*L281: ADP (10 µm) -> ADP (10 µM)

*L341: Figure 8E and 8F: pretease E vs. pronase E

*L373: Figure 9C: the label of y-axis TLR4

*L434: including cytokines, cytokines?,

L477, L483: CO2

L481: 5.0×103

L495: phorbol 12-myristate 13-acetate (PMA) -> give the source

L499: RUX

L510: 1.0 × 106

L535: TPO full name and the source

*L538: the mice in the control group and model group were intragastrically administered normal saline per day -> why?

554: 100×104

567: Na2HPO4

568: HCO3, MgCl2, 1 mM CaCl2

572: ADP source

578: minutes vs. min

584: h vs hour

598: 4.16 Protein–Protein Interaction (PPI) Network Construction and Screening of Core Targets

621: 4.18 Drug Affinity Responsive Target Stabilization Assay (DARTS)

*660: ?% SDS-PAGE

L740: Tubulin in Platelets: When the Shape Matters.

L859: mice. . Genes Dev.

Author Response

Dec. 13, 2022

Dear Expert Reviewer,

Thank you very much for the prompt review process and excellent comments. We greatly appreciate the time and efforts which you have spent on it. We are submitting the revised manuscript entitled “Targeting TLR2/Rac1/cdc42/JNK pathway to reveal that ruxolitinib promotes thrombocytopoiesis” (ID: ijms-2075065) to International Journal of Molecular Sciences.

We have carefully considered your comments and suggestions, and addressed each of the concerns in response to the comments (see point by point response). We have revised the manuscripts based on your comments and carefully checked throughout the manuscript and corrected the language errors. Our point-by-point responses to the comments (in blue) are shown below (in red).

Ruxolitinib is used as a selective JAK1/2 inhibitor with moderate tyrosine kinase inhibitory activity (Tyk2). The present study demonstrated that Ruxolitinib could act as a TLR2 agonist to activate Rac1/cdc42/JNK pathway, leading to stimulate megakaryocyte differentiation and accelerate recovery of megakaryocytes and thrombocytopoiesis in a mouse model of radiation-injured thrombocytopenia (RIT).  There are some concerns as listed in the following:

(1) To consider the drug target specificity and the side effect, the authors should give any comments on the effective concentration for TLR2 activation vs. JAK1/2 inhibition as well as the relationship between TLR2-mediated signaling pathway and JAK1/2.

Response: Thank you very much for the excellent comments. As suggested, we have added the discussion about this point in discussion section (page 15, line 496-519).

 (2) The result statements of 2.7 did not match the Figure 9.

Response: Thank you for your careful reading and kind reminding. We are sorry that we did not express the order of results correctly, and in the revised version, we have modified it (page 13, line 388-390).

(3) Typos and others:

L31: TLR2 full name first

Response: We have added the full name of TLR2 in the revised manuscript (page 1, line 31).

L73: differentiation survival and apoptosis

Response: Thank you for your thoughtful comments. we have modified it (page 2, line 72).

L126: * p <.05

Response: We have made correction in revised manuscript (page 4, line 136).

L162: TPO full name first

Response: We have added the full name of TPO in revised manuscript (page 5, line 172).

*L176: change the sequence of TPO label and Model label in Figure 3B

Response: Thanks a lot for the careful suggestion. In the revised manuscript, we have adjusted the sequence of TPO label and model label in new Figure 3B.

*L214, L239: at magnifications of 200× (top) and 100× (bottom)?

Response: We have corrected it in the revised manuscript (page 7, Figure 226; page 8, line 252).

L272: Figure 6G

Response: Thank you for your careful check! In revised manuscript, we have uploaded the Figure 6.

*L281: ADP (10 µm) -> ADP (10 µM)

Response: We are very sorry for the omission or miswriting of the concentration unit on the same page and we have uniformly modified the concentration unit to be µM in the revised version (page 9, line 295).

*L341: Figure 8E and 8F: pretease E vs. pronase E

Response: We have changed " pretease E " to the correct format " pronase E " in  new Figure 8.

*L373: Figure 9C: the label of y-axis TLR4

Response: We have made corresponding modification in revised manuscript.

*L434: including cytokines, cytokines?,

Response: Thank you for your careful reading. In the revised version, we have removed the duplicate content (page 15, line 460).

L477, L483: CO2

Response: This section has been updated in the revised manuscript (page 16, line 536; 542).

L481: 5.0×103

Response: We have made the appropriate modifications in the revised manuscript (page 16, line 540).

L495: phorbol 12-myristate 13-acetate (PMA) -> give the source

Response: Thanks for your careful reading. According to your suggestions, we have added the source of PMA in the revised manuscript (page 16, line 553).

L499: RUX

Response: In the revised manuscript, we changed “RUX” to “ruxolitinib” (page 17, line 562).

L510: 1.0 × 106

Response: In the revised manuscript, we changed “100×104” to “100×104” (page 17, line 572).

L535: TPO full name and the source

Response: We have added the full name and source of TPO in the revised manuscript (page 5, line 172) (page 17, line 597).

*L538: the mice in the control group and model group were intragastrically administered normal saline per day -> why?

Response: Thank you for your rigorous thinking. We are very sorry that in this section we mistakenly used “intragastrical administered”, which should correctly be “intraperitoneal injection”. In the revised manuscript, we have modified it (page 17, line 600). In the in vivo pharmacodynamic study, rhTPO and ruxolitinib were diluted to 10 g/0.1 ml with saline, and then injected intraperitoneally according to the weight of the mice. To control for variables, we also performed the same operation on the control mice and model mice, taking intraperitoneal injections of 10g/0.1ml saline daily according to the body weight of the mice.

554: 100×104

Response: In the revised manuscript, we changed “100×104” to “100×104” (page 17, line 615).

567: Na2HPO4

Response: This section has been updated in the revised manuscript (page 18, line 631).

568: HCO3, MgCl2, 1 mM CaCl2

Response: In the revised manuscript, this section has been updated (page 18, line 632).

572: ADP source

Response: We have supplemented this section in the revised manuscript (page 18, line 636).

578: minutes vs. min

Response: We changed “minutes” to “min” (page 18, line 643; 644).

584: h vs hour

Response: We changed “h” to “hour” (page 18, line 648).

598: 4.16 Protein–Protein Interaction (PPI) Network Construction and Screening of Core Targets

Response: Thank you for the reminder. We have made the corresponding changes in the revised manuscript (page 18, line 662).

621: 4.18 Drug Affinity Responsive Target Stabilization Assay (DARTS)

Response: The corresponding changes have been made in the revised manuscript (page 19, line 695).

*660: ?% SDS-PAGE

Response: We have supplemented this section in the revised manuscript (page 19, line 734).

L740: Tubulin in Platelets: When the Shape Matters.

Response: We have checked all the references and corrected the errors in revised manuscript.

L859: mice. . Genes Dev.

Response: Thanks for your careful review. We have updated the references.

Thank you for all the valuable and helpful comments and suggestions.

Best regards,

Jianming Wu

Round 2

Reviewer 1 Report

The revised version is acceptable for publication.